# Austalide K from the Fungus *Penicillium rudallense* Prevents LPS-Induced Bone Loss in Mice by Inhibiting Osteoclast Differentiation and Promoting Osteoblast Differentiation

**DOI:** 10.3390/ijms22115493

**Published:** 2021-05-23

**Authors:** Kwang-Jin Kim, Jusung Lee, Weihong Wang, Yongjin Lee, Eunseok Oh, Kyu-Hyung Park, Chanyoon Park, Gee-Eun Woo, Young-Jin Son, Heonjoong Kang

**Affiliations:** 1Department of Pharmacy, Sunchon National University, 315 Maegok-dong, Suncheon 57922, Korea; mastiffk@naver.com (K.-J.K.); yojilee@gmail.com (Y.L.); 2Laboratory of Marine Drugs, School of Earth and Environmental Sciences, Seoul National University, NS-80, Seoul 08826, Korea; leejusung@snu.ac.kr (J.L.); pharmacy2007@naver.com (W.W.); dmstjr0130@snu.ac.kr (E.O.); grekhp@snu.ac.kr (K.-H.P.); tddok7953@naver.com (G.-E.W.); 3Interdisciplinary Graduate Program in Genetic Engineering, Seoul National University, NS-80, Seoul 08826, Korea; chanyoon0126@snu.ac.kr; 4Research Institute of Oceanography, Seoul National University, NS-80, Seoul 08826, Korea

**Keywords:** marine fungus, osteoporosis, bone diseases, bone remodeling

## Abstract

Osteoporosis is a chronic disease that has become a serious public health problem due to the associated reduction in quality of life and its increasing financial burden. It is known that inhibiting osteoclast differentiation and promoting osteoblast formation prevents osteoporosis. As there is no drug with this dual activity without clinical side effects, new alternatives are needed. Here, we demonstrate that austalide K, isolated from the marine fungus *Penicillium rudallenes*, has dual activities in bone remodeling. Austalide K inhibits the receptor activator of nuclear factor-κB ligand (RANKL)-induced osteoclast differentiation and improves bone morphogenetic protein (BMP)-2-mediated osteoblast differentiation in vitro without cytotoxicity. The nuclear factor of activated T cells c1 (NFATc1), tartrate-resistant acid phosphatase (TRAP), dendritic cell-specific transmembrane protein (DC-STAMP), and cathepsin K (CTSK) osteoclast-formation-related genes were reduced and alkaline phosphatase (ALP), runt-related transcription factor 2 (Runx2), osteocalcin (OCN), and osteopontin (OPN) (osteoblast activation-related genes) were simultaneously upregulated by treatment with austalide K. Furthermore, austalide K showed good efficacy in an LPS-induced bone loss in vivo model. Bone volume, trabecular separation, trabecular thickness, and bone mineral density were recovered by austalide K. On the basis of these results, austalide K may lead to new drug treatments for bone diseases such as osteoporosis.

## 1. Introduction

Osteoporosis is an age- and sex-related disease for which postmenopausal women are at the greatest risk [1]. In addition to postmenopausal osteoporosis, bone remodeling plays an important role in rheumatoid arthritis, periodontitis, and Paget’s disease. Bone remodeling is a process where bone is repeatedly produced and destroyed (resorbed) through active metabolism, and it mainly relies on the activity of osteoclasts and osteoblasts [2]. Furthermore, 20% of total bone is replaced by new tissue every year [3].

The process of bone remodeling can be described as having four stages. First, osteoclasts derived from hematopoietic stem cells are activated for resorption from dormant bones covered with inactive osteoblasts by cytokines such as RANKL and M-CSF. Second, the activated mature osteoclasts are fused and form multinucleated osteoclasts via DC-STAMP, osteoclast stimulatory transmembrane protein (OC-STAMP), and v-ATPase V0 subunit d2 (Atp6v0d2). The multinucleated osteoclasts attach to the bone surface via the avb3 integrin receptor and produce protons with the help of carbonic anhydrase II (CA II) and vacuolar H^+^ -ATPase. This resorption process lasts about 3 weeks, using the chloride channel-7 (CLC7), protease CTSK enzyme, and acid. Third, after the resorption process and following entry into the recovery phase, immature osteoblasts differentiate into mature osteoblasts. BMPs, a TGF-β superfamily, regulate signaling in immature osteoblasts, and mediate osteoblast maturation by expressing signaling agents that regulate Wnt protein interactions between cells. In addition, IGF signaling, which controls the differentiation of parathyroid hormone, growth hormone, and osteoblasts, also plays an important role, resulting in osteoblasts forming a collagenous bone matrix and subsequently in mineralization. Lastly, bone matrix proteins such as collagen type I, osteocalcin, and alkaline phosphatase are produced and cause osteoblasts to form new bone for several months. If the balance between osteoclast and osteoblast activity is disrupted, it leads to an increased possibility of bone fractures [4,5,6,7].

Various drugs to treat or prevent osteoporosis are in clinical use, falling mostly into two categories: (1) inhibitors of bone reabsorption and (2) accelerators of bone formation [8]. Bisphosphonates (osteoclast-targeting), denosumab (anti-RANKL monoclonal antibody, osteoclast-targeting), and teriparatide (osteoblast-targeting) have been proposed as first-line therapies in osteoporosis treatment [9]. Calcitonin (osteoclast-targeting), selective estrogen receptor modulator (SERM, osteoclast-targeting), sodium fluoride (osteoblast-targeting), parathyroid hormone (osteoblast-targeting), and strontium ranelate (osteoclast and osteoblast targeting) are sometimes used according to the patient’s age and the severity of the condition [9]. Bisphosphonates, powerful bone resorption inhibitors, have a number of advantages including a distinct fracture-prevention effect, increased bone density, and cost-effective economics; however, when used for a long time, the bone conversion rate (bone replacement rate) decreases and side effects such as accumulation of microdamage and jaw bone necrosis may occur [10,11]. Typically, bisphosphonates have adverse effects such as dyspepsia, abdominal pain, and rash [12]. Denosumab is associated with dermatitis, rash, and mild muscle pain. In the case of teriparatide, side effects such as transient hypercalcemia, nausea, and rhinitis are common [13].

Strontium ranelate is first drug to be approved in Europe that has been promoted as a dual-action agent (increase of new bone formation and reduction of bone resorption). Even though it has significant efficacy against osteoporosis, the European Pharmacovigilance Risk Assessment Committee has recommended its limited use due to associated heart problems [14]. For this reason, the US FDA has not approved the drug for use in the United States [15,16]. Therefore, new and effective treatment agents for osteoporosis are urgently required.

The investigation of natural products to meet the need for treatment of osteoporosis is steadily underway. In previous studies, our group discovered several new drug leads for osteoporosis treatment [17,18]. In particular, in the last two decades, marine-derived natural products have attracted more and more attention from natural product chemists and pharmacologists because of their diverse structures and profound biological activities. To extend our continuing efforts, we aimed to discover bioactives with pro-osteogenic and anti-osteoclastogenic dual activities. Using bioactivity-guided fractionation, we discovered a dual-functional natural product that can be isolated from the marine fungus *Penicillium rudallense*. Austalide K, belonging to the meroterpenoid class, was first isolated from a culture of *Aspergillus ustus*. To date, as many as 34 members of the austalide family have been isolated, mainly from marine-derived *Aspergillus* and *Penicillium* species. Only a few of them have been reported to have biological activities, including antibacterial and endo-1, 3-β-d-glucanase-inhibitory activities [19,20]. Our previous study revealed the osteoclast-differentiation-inhibitory activity of austalides V, W, L, and P, 17S-dihydroaustalide K, and deacetoxyaustalide I, which prompted further biological study of other analogues. In this study, we describe the biological activities of austalide K (see Appendix A) on bone metabolism in vitro and in vivo, including osteoclast-inhibitory activity and osteoblast-promotional activity.

## 2. Results

### 2.1. Chemical Structure of Austalide K

Austalide K was obtained via bioactivity-guided fractionation. In brief, after fermentation in a shaking incubator for 9 days, the fungal mycelia were separated from the broth by filtration using cheese cloth. The filtered mycelia were extracted with methanol (MtOH) and acetone (1:1) and the culture broth was partitioned with ethyl acetate. The two organic extracts were combined and fractionated by MPLC into eight fractions using a step-gradient solvent system. Fraction 2 was active in the bioassay and was subjected to a semipreparative reversed-phase HPLC (Phenomenex Luna C18 (2), 5 μm, 100 Å, 250 × 100 mm, 2.0 mL/min, UV = 210 nm), followed by elution with 68% acetonitrile in water to afford austalide K (retention time: 39 min).

Austalide K has a molecular formula of C_25_H_32_O_5_, which was determined from the high-resolution ESIMS data of the protonated molecule peak at m/z 413.2320 [M + H]+. The UV spectrum showed maximum absorptions at 222 and 268 nm, which is reminiscent of a substituted phthalide moiety. The 1H NMR spectrum (Appendix A) indicated the presence of one aromatic methyl group, four aliphatic methyl groups, and one methoxy group. In addition, the 1H NMR spectrum showed signals assigned to six methylenes and two methines, including an oxygenated benzylic methylene proton signal at δ 5.12. Combined analysis of the ^1^H, ^13^C, and 2D NMR data (HSQC, COSY, and HMBC) established the pentacyclic structure. The relative configuration of austalide K was revealed by analysis of the ROESY spectra. The NOE correlation between H3-24 and H-21 suggested the cis-fusion relationship of rings C and D. Ring D was trans-fused with ring E, which was inferred from the NOE correlations H3-26/H3-27 and H-14/H3-25. The NOE correlations H3-24/H-22b (δH 2.81) and H3-27/H-22a (δH 2.93) indicated a trans relationship between H3-24 and H3-27. The absolute configuration was assigned by comparison of the value of optical rotation with that previously reported (Figure 1).

### 2.2. Effect of Austalide K on RANKL-Induced Osteoclast Differentiation

Austalide K was found to suppress RANKL-induced osteoclast differentiation without cytotoxicity. BMMs were incubated with austalide K to identify its effect on RANKL-induced osteoclast differentiation. TRAP staining was used to detect the number of mature osteoclasts, as TRAP is known to be highly expressed in osteoclasts [21]. Austalide K inhibited TRAP-positive cells in a dose-dependent manner and had no cytotoxicity up to 20 μM (Figure 2A,C). The significant TRAP-inhibitory effect was also confirmed by evaluation of mRNA expression of TRAP (Figure 2B). Collectively, these results suggest that austalide K exhibits an inhibitory effect on RANKL-induced osteoclast differentiation in BMMs.

### 2.3. Effect of Austalide K on Expression of Osteoclastogenesis-Related Genes.

To further confirm the inhibitory effect on osteoclast differentiation, osteoclastogenesis-related genes were examined. Austalide K attenuated mRNA and protein expression of *NFATc1*, which is one of the important transcription factors involved in the differentiation of BMMs into osteoclasts (Figure 3A,B). From the results, we confirmed the effect of austalide K in suppressing RANKL-induced osteoclast differentiation, reducing *NFATc1* expression and protein levels, and suppressing related target genes.

### 2.4. Austalide K Increased Osteoblast Differentiation

Austalide K was further found to increase BMP-2-induced osteoblast differentiation without toxicity. To confirm the effect of austalide K on BMP-2-induced osteogenesis, C2C12 cells were incubated with austalide K in various concentrations, followed by BMP-2 (100 ng/mL). Austalide K enhanced the mRNA expression of *ALP* (Figure 4A), a marker of osteogenic differentiation, in a dose-dependent manner (Figure 4A, B) without affecting the viability of the cells (Figure 4C).

### 2.5. Austalide K Increased Osteoblast-Differentiation-Related Genes

During the differentiation of the C2C12 cells, levels of osteogenesis marker genes such as *Runx2, OCN*, and *OPN* showed elevated expression after treatment with austalide K (Figure 5). These results suggest that austalide K promotes osteogenic differentiation by activating *Runx2* and its target genes *OCN,* and *OPN* in osteoblasts in vitro.

### 2.6. Austalide K Changed Bone Parameters of LPS-Induced Bone Loss in In Vivo Model

To demonstrate the preventive effect of austalide K on an in vivo LPS-induced bone loss model, mice were treated with PBS, LPS, or LPS + austalide K for 7 days (LPS: Day 1 and 4, austalide K: daily). Micro-CT analysis indicated that austalide K treatment protected against LPS-induced bone loss in a dose-dependent manner (Figure 6A). Furthermore, austalide K significantly prevented changes in bone mineral density, bone volume, trabecular thickness, and trabecular separation number induced by LPS (Figure 6B). At 4 mg/kg concentration, bone mineral density was improved 2.25-fold, bone volume 2.9-folds, trabecular thickness 1.16-fold, and trabecular separation number 1.62-fold. Our data clearly demonstrated that treatment with austalide K protected against LPS-induced bone loss in vivo.

## 3. Discussion

Numerous natural products have been reported to exhibit anti-osteoporotic activity, but active compounds that inhibit osteoclasts and promote osteoblasts at the same time are rare. In particular, few anti-osteoporotic compounds with dual action except acredinone have been derived from microorganisms [18]. In the present study, we identified the novel activity on bone metabolism of austalide K, isolated from cultures of the marine fungus *Penicillium rudallense*, by demonstrating the simultaneous regulation of the genes related to osteoclast and osteoblast differentiation, as in the acredinone study [14].

Austalide K suppressed RANKL-induced osteoclast differentiation by inhibiting NFATc1 expression, which led to downregulation of its target genes, *TRAP, DC-STAMP*, and *CTSK*. In line with the TRAP-staining assay (Figure 2A), mRNA expression of *TRAP* was inhibited by austalide K treatment (Figure 3A). Furthermore, CTSK, a protease that degrades collagen and matrix proteins during bone resorption [22], was also inhibited in a dose-dependent manner. DC-STAMP is known to be involved in cell–cell fusion, which is essential for the generation of intact multinucleated osteoclasts [23]. As shown in Figure 3A, austalide K treatment significantly suppressed mRNA expression of *DC-STAMP*. These results demonstrate that austalide K inhibited RANKL-induced osteoclast differentiation by diminishing *NFATc1* mRNA expression and protein levels, and suppressing its related target genes. These are essential genes for the generation of mature multinucleated osteoclasts [22,23,24]. In addition, the average percentage reduction of gene expression of the osteoclatogenic major factors by austalide K was more than 80%, compared to around 50% by acredinone [14]. This means that austalide K strongly inhibited the expression of osteoclast-related genes and might be used as a scaffold for much more effective osteoporotic treatment medicines.

Austalide K also activated *Runx2*, which is required for the proliferation of osteoblastic progenitors, thereby upregulating its target genes *OCN, OPN*, and *ALP* to activate osteoblast differentiation [25,26]. As Runx2 is the specific transcription factor of osteoblasts [27,28], it controls the osteogenic differentiation process by regulating OCN, OPN, and ALP. When we compared austalide K and acredinone in the induction of osteoblast activity, acredinone had a greater effect on the gene expression of *ALP*, whereas austalide K had a greater effect on the gene expression of *OCN* and *OPN*. The osteoblastic activities of austalide K and acredinone may be similar, according to some experimental results.

Thus, we confirmed that austalide K can not only inhibit osteoclast differentiation and but also activate osteoblast differentiation. Compared with the previous results of the acredinone study, austalide K had a greater activity on the inhibition of osteoclast differentiation, as well as on the formation of osteoblasts at a concentration of 10 μM. Finally, we performed a bone erosion assay. Austalide K protected against LPS-induced bone loss in mice, resulting in the improvement of bone mineral density, bone volume, trabecular thickness, and trabecular separation number. Furthermore, pharmacokinetics plays an important role in the progress of drug development [29]. Austalide K did not present satisfactory bioavailability, although it showed profound in vivo activity. Characterization and biological activity assessment of the major metabolites of austalide K is in progress. Taken together, austalide K may lead to new therapeutic drugs for bone-loss-related diseases.

The ocean has a rich biological diversity of marine microorganisms due to the harsh and stressful conditions, including cold, high pressure, salt, and darkness. These produce diverse compounds with unique chemical structures and good biological activities. With recent advances in molecular genetic techniques, the shortcomings of classical marine natural product studies (low fermentation yields) are being overcome. Therefore, marine-microorganism-derived natural products are being actively researched as attractive resources for innovation in the drug, cosmetic, food, and other industries [30,31].

## 4. Materials and Methods

### 4.1. General Experimental Procedures

UV spectra were obtained with a Hitachi JP/U-3010 UV spectrophotometer (Tokyo, Japan). IR spectra were acquired on a JASCO FT/IR 4200 spectrophotometer (Tokyo, Japan). All NMR spectra were recorded on a Bruker AscendTM 700 spectrometer (Billerica Middlesex County, MA, USA). Electrospray ionization source (ESI) low-resolution mass spectra were recorded on an Agilent Technologies 6120 quadrupole mass spectrometer coupled with an Agilent Technologies 1260 series HPLC (Santa Clara, CA, USA). High-resolution mass spectrometric data were collected on a JEOL JMS-700 double-focusing (B/E configuration) instrument (Tokyo, Japan). High-performance liquid chromatography was performed with a HPLC of Waters Corporation (Milford, MA, USA) equipped with a Waters 2998 photodiode array detector and Waters 1525 binary pump (Milford Worcester County, MA, USA). HPLC-grade solvents from Burdick & Jackson (Charlotte, NC, USA) were used for HPLC. NMR solvents were purchased from Cambridge Isotope Laboratories (CIL) Inc. (Andover, MA, USA).

### 4.2. Isolation and Cultivation of the Fungal Strain

The fungal strain *Penicillium rudallense* was isolated from marine sediment collected on Ga-geo Island, Republic of Korea. The fungus was identified using internal transcribed spacer (ITS) region sequencing. A BLAST search showed the highest similarity to *Penicillium rudallense*. The sequence data of ITS 1 and 4 of the fungus were deposited in GenBank with accession numbers MK209625 and MK409626, respectively.

The marine fungus was cultured on potato dextrose agar (PDB, Difco) using natural seawater. The cultured fungus agar blocks were inoculated into plastic culture flasks containing 1 L of PDB, and then incubated under shaking conditions (110 rpm) at 25 °C for 7 days.

### 4.3. Extraction and Isolation of Secondary Metabolites from Seawater PDB Medium

The fungal mycelium was separated from the broth (4 L) by filtration. The filtered culture broth was partitioned with ethyl acetate (EtOAc). The filtered mycelium was extracted with methanol (MtOH) and acetone (1:1) and filtered. The filtrate from the mycelium was evaporated and partitioned with EtOAc. The two EtOAc extracts (2.03 g) were combined and subjected to silica gel column chromatography using a step-gradient solvent system consisting of dichloromethane and MtOH, which yielded six fractions. Fraction 2 was purified on a reversed-phase HPLC column (Phenomenex Luna C18 (2), 5 μm, 100 Å, 250 × 100 mm, 2.0 mL/min, UV = 210 nm) and eluted with 60% acetonitrile (CH_3_CN), and austalide K was obtained (40.1 mg).

Austalide K: white amorphous powder; [α]^20^_D_ = −75.9 (*c* 0.40, MtOH); UV (MtOH) *λ_max_* (*log ε*) 222 (4.18), 268 (3.82); ECD (4.8 × 10^−3^ M, MtOH) *λmax* (∆ε) 266 nm (−5.26), 230 nm (+3.78), 213 nm (−7.43); IR (KBr) *ν_max_* 2932, 1749, 1702, 1609, 1476, 1458, 1436, 1366, 1312, 1283, 1139, 1070, 1046, 986, 903 cm^−1^; positive ESIMS *m/z* 413 [M + H]^+^, 435 [M + Na]^+^; HRESIMS *m/z* 413.2330 [M + H]^+^ (calcd for C_25_H_33_O_5_, 413.2328).

### 4.4. Osteoclast Cell Culture and Differentiation

Bone-marrow-derived macrophage (BMM) culture was performed rigorously according to the recommendations contained in the Standard Protocol for Animal Study of Sunchon National University. All cells were cultured at 37 °C in a 5% CO_2_ incubator. The BMCs were isolated from the tibia and femur of 5 week old male ICR mice (*n* = 2: Damool Science, KR) by flushing with α-minimum essential medium (α-MEM; Invitrogen Life Technologies, Carlsbad, CA, USA) supplemented with 100 U/mL penicillin/streptomycin (Invitrogen, Carlsbad, CA, USA). The cells were incubated on a Petri dish in α-MEM supplemented with 10% fetal bovine serum (FBS; Invitrogen Life Technologies, Carlsbad, CA, USA) and 100 U/mL penicillin/streptomycin (10% α-MEM) with 30 ng/mL of the mouse recombinant macrophage colony-stimulating factor (M-CSF; PEPROTECH, NJ, USA). After 3 days, the cells attached to Petri dishes were obtained as BMMs.

For the osteoclast-differentiation experiment, M-CSF (30 ng/mL; Peprotech, Rochky Hill, NJ, USA) and RANKL (10 ng /mL; R&D Systems, Minneapolis, MN, USA) were added to BMMs and cultured for 4 days.

### 4.5. TRAP-Staining Assay

TRAP, a biomarker of osteoclast differentiation, was used as a stain to visualize mature osteoclasts. Cells were fixed with 3.7% formalin for 10 min, permeabilized with 0.1% Triton X-100 for 10 min, and stained with TRAP solution.

### 4.6. Osteoclast Cytotoxicity Assay

BMMs were plated in triplicate at a density of 1 × 10^4^ cells/well on 96-well plates. After treatment with M-CSF (30 ng/mL) and austalide K, cells were incubated for 3 days. Cell viability was measured using the Cell Counting Kit-8 (CCK-8; Dojindo Molecular Technologies, Kumamoto, Japan) according to the manufacturer’s protocol.

### 4.7. Osteoblast Culture and Differentiation

Mouse mesenchymal precursor C2C12 cells were purchased from American Type Culture Collection and maintained in Dulbecco’s modified Eagle’s medium (DMEM) containing 10% FBS, 100 U/mL penicillin, and 100 μg/mL streptomycin. Cells were seeded in 96-well plates or in 6-well plates and, after 1 day, cells were differentiated by replacing the medium with DMEM containing 5% FBS and rhBMP-2 (50 ng/mL).

### 4.8. Alkaline Phosphatase Staining

C2C12 cells were plated in triplicate at a density of 8 × 10^3^ cells/well on 96-well plates. After treatment with BMP-2 (100 ng/mL; R&D Systems, Minneapolis, MN, USA) and austalide K, the cells were cultured for 4 days, with the medium replaced with fresh medium after 3 days. The cells were then fixed with 10% formalin in PBS for 1 min and washed with deionized water. An ALP-staining kit was used in the dark. A DP60 digital camera was used to obtain the images.

### 4.9. Osteoblast Cytotoxicity Assay

C2C12 cells were plated in triplicate at 8 × 10^3^ cells/well on 96-well plates. After treatment with BMP-2 (100 ng/mL) and austalide K, cells were cultured for 4 days. A CCK-8 kit was used to measure cell viability according to the manufacturer’s protocol.

### 4.10. Western Blot and Real-Time PCR Analysis

Western blot and real-time PCR analysis were performed as described previously [17]. Antibodies against NFATc1 and actin were purchased from Santa Cruz Biotechnology (CA, USA). Primers (Table 1) were chosen with the online Primer3 design program [32].

### 4.11. In Vivo LPS-Induced Bone Erosion Model

Male ICR mice (5 weeks old) were divided into four groups of five mice. LPS was intraperitoneally injected to induce bone erosion. Mice were injected intraperitoneally with austalide K or 5% Kolliphor EL in PBS (vehicle) daily for 7 days. On the 8th day after austalide K administration, the mice were sacrificed by cervical dislocation and the femurs were obtained. The femurs were analyzed by obtaining images and data using micro-CT. The resulting femurs were fixed with 4% paraformaldehyde for 1 day and then demineralized with 12% EDTA. High-resolution micro-CT (SkyScan 1272) and DataViewer (SkyScan) were used to scan femurs. The bone mineral density (BMD), bone volume/total volume (BV/TV), bone surface/total volume (BS/TV), and trabecular separation (Tb. Sp) were measured to assess the trabecular bone microstructure of the femurs, using the CTAn software provided with the SKYSCAN analysis tool. The experimental protocol was approved by the Sunchon National University Institutional Animal Care and Use Committee (SCNUIACUC; Permit No. SCNU IACUC 2016-06).

### 4.12. Statistical Analysis

All quantitative values are presented as mean ± standard deviation. Statistical differences were analyzed using Student’s *t*-test. Value of *p* < 0.05 were considered significant, and is indicated as * *p* < 0.05, ** *p* < 0.001, *** *p* < 0.0001 vs. vehicle.

## 5. Conclusions

In conclusion, we demonstrated for the first time that the marine microbial metabolite austalide K, isolated from the marine fungus *Penicillium rudallense*, has unique activity on bone remodeling via the inhibition of osteoclastogenesis and promotion of osteoblast formation. This activity was confirmed by analyzing the expression of markers associated with the bone remodeling process. Austalide K also protected against the LPS-induced bone loss in mice. Therefore, austalide K may lead to novel drugs for osteoporosis. Further studies for identifying cellular targets are in progress.

## Figures and Tables

**Figure 1 ijms-22-05493-f001:**
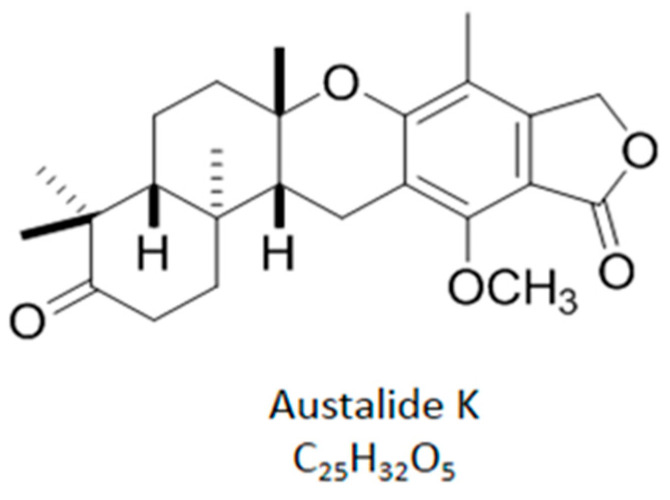
Chemical structure of austalide K from *Penicillium rudallens*.

**Figure 2 ijms-22-05493-f002:**
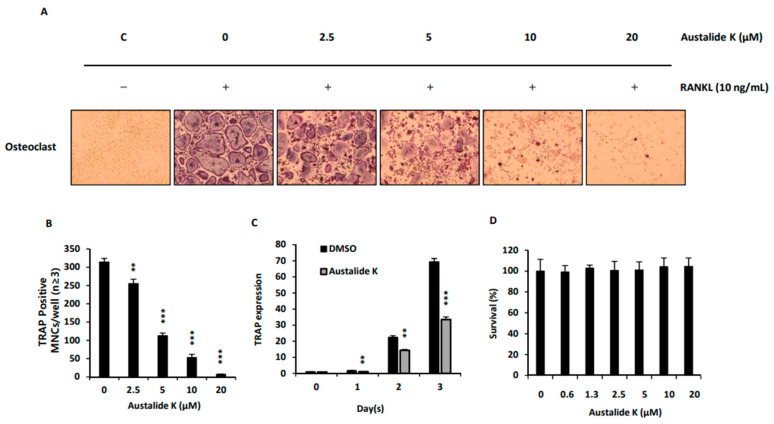
Effect of Austalide K on RANKL-induced osteoclast differentiation. (**A**) BMMs were incubated in 96-well plates containing 1 × 10^4^ cells/well. After a day, RANKL (10 ng/mL) and M-CSF (30 ng/mL) were treated with austalide K for 4 days. Cells were TRAP-stained with TRAP solution; (**B**) TRAP-positive multinucleated cells (3 or more nuclei) were counted as osteoclasts. (**C**) Relative TRAP mRNA expression levels were evaluated using qRT-PCR; (**D**) Cytotoxicity of austalide K was evaluated by CCK-8. Cells were incubated on 96-well plates containing 1 × 10^4^ cells/well and treated with austalide K for 3 days. ** *p* < 0.01; *** *p* < 0.001. (*n* = 3).

**Figure 3 ijms-22-05493-f003:**
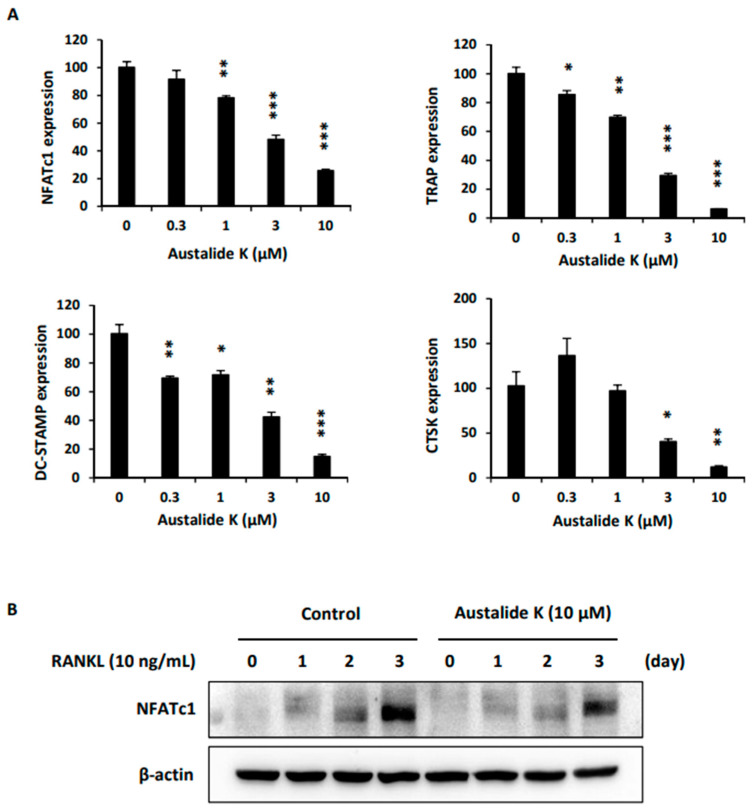
Austalide K suppressed NFATc1 in RANKL-treated osteoclasts. (**A**) The mRNA expression levels of *NFATc1, TRAP, DC-STAMP*, and *CTSK* were detected using qRT-PCR at 3 days after RANKL treatment; (**B**) The expression level of NFATc1 was measured using Western blot analysis. * *p* < 0.05; ** *p* < 0.01; *** *p* < 0.001.

**Figure 4 ijms-22-05493-f004:**
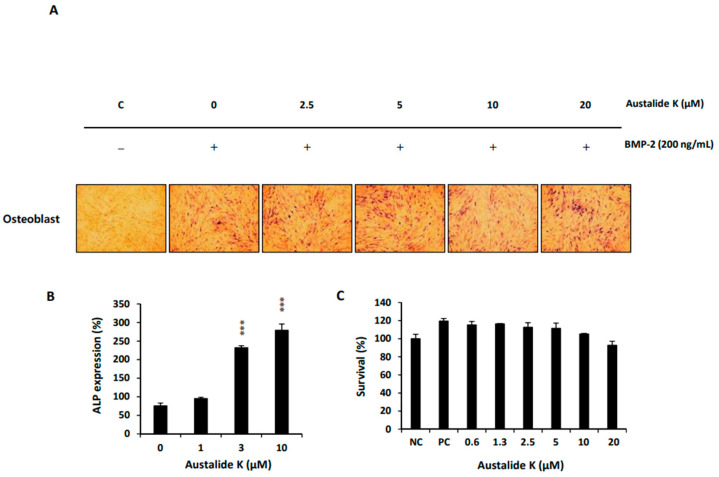
Austalide K increased osteoblast differentiation. (**A**) Austalide K stimulated BMP-2-induced (200 ng/mL) ALP expression for 4 days; (**B**) Relative ALP mRNA expression levels were evaluated by qRT-PCR; (**C**) Cytotoxicity of austalide K in C2C12 cells was measured using a CCK-8 kit. C2C12 cells were plated on 96-well plates at 8 × 10^3^ cells/well and treated with austalide K for 4 days. NC (negative control) and PC (positive control) were BMP-2-untreated and -treated C2C12 cells, respectively. *** *p* < 0.001.

**Figure 5 ijms-22-05493-f005:**
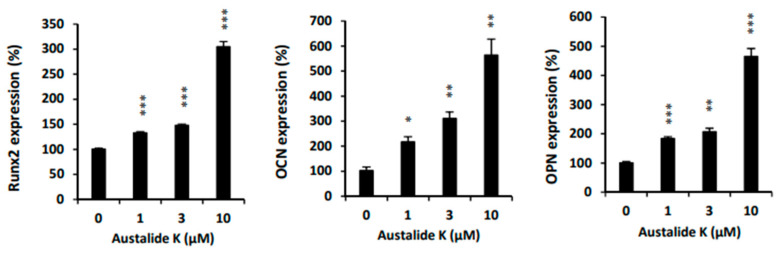
Austalide K increased osteoblast-differentiation-related genes. Relative *Runx2, OCN*, and *OPN* mRNA expression levels were evaluated by qRT-PCR after 4 days. C2C12 cells were plated in 6-well plates at 8 × 10^4^ cells/well and were treated with austalide K for 6 days. * *p* < 0.05; ** *p* < 0.01; *** *p* < 0.001.

**Figure 6 ijms-22-05493-f006:**
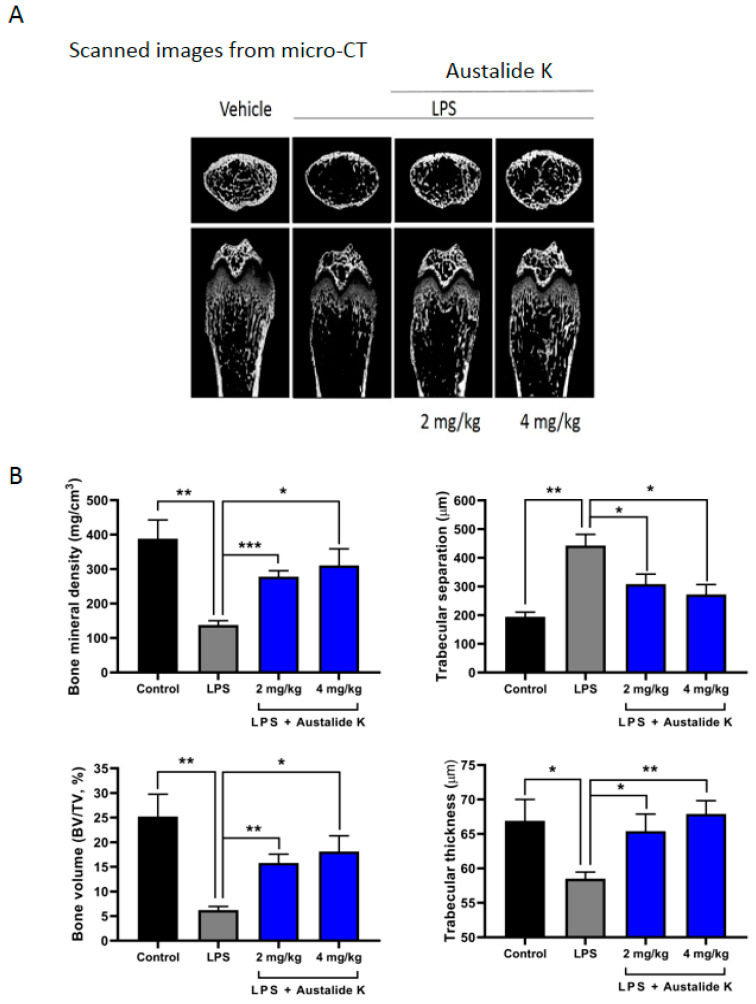
Austalide K changed bone parameters of LPS-induced bone loss in in vivo model. (**A**) Transverse and longitudinal micro-CT images of austalide K-treated mice; (**B**) Bone mineral density, trabecular separation, bone volume, and trabecular thickness were measured. * *p* < 0.05; ** *p* < 0.01; *** *p* < 0.001.

**Table 1 ijms-22-05493-t001:** Primer sequences used in this study.

Target Gene	Forward Primer (5′–3′)	Reverse Primer (5′–3′)
TRAP	GATGACTTTGCCAGTCAGCA	ACATAGCCCACACCGTTCTC
NFATc1	GGGTCAGTGTGACCGAAGAT	GGAAGTCAGAAGTGGGTGGA
DC-STAMP	CCAAGGAGTCGTCCATGATT	GGCTGCTTTGATCGTTTCTC
CTSK	GGCCAACTCAAGAAGAAAAC	GTGCTTGCTTCCCTTCTGG
ALP	GATGGCGTATGCCTCCTGCA	CGGTGGTGGGCCACAAAAGG
Runx2	CCAGAATGATGGTGTTGACG	AGGGTTGCAAGATCATGACT
OCN	TATGTGTCCTCCGGGTTCAT	GCCCTCTGCAGGTCATAGAG
OPN	GCTAGTCCTAGACCCTAAGA	TCCTGCTTAATCCTCACTAA
GAPDH	AACTTTGGCATTGTGGAAGG	ACACATTGGGGGTAGGAACA

## Data Availability

The NMR data are available online.

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
