# Peer review of "Austalide K from the Fungus Penicillium rudallense Prevents LPS-Induced Bone Loss in Mice by Inhibiting Osteoclast Differentiation and Promoting Osteoblast Differentiation"

_ijms, 2021, doi:10.3390/ijms22115493_

Round 1

Reviewer 1 Report

This is an incredible research article in bone research and may open new hope for the treatment of bone diseases by targeting bone resorption and formation simultaneously, which may lead to potential drug development. This research and manuscript design is very comprehensive and appropriate for the International Journal of Molecular Sciences. However, I have the following suggestions: 1. Please increase the size of Figure 2A, 4A and mark the cells for clarity. 2. Line 128, levels of NFATc1, TRAP. DC-STAMP; please replace with comma. 3. Figure 4C, please keep the X-axis name at the middle of the corresponding column bar. 4. Figure 5 (middle one), XY-axis color is different. 5. Please use italic wherever required, for example, line 158, in vivo, etc. 6. Please use the appropriate symbols, for example, line 23, nuclear factor-kB ligand. It should be κ. 7. Line 181, inhibiting NFAT c1 expression; please delete the space. 8. Line 195, Taken together, austalide K may lead to novel drugs for treating osteoporosis. I think it’s better to write may lead to the development or discovery or something like that. Also, check line 304. 9. In the material and method section, please mention how Bone marrow-derived macrophages cells were collected, separated. 10. 4.6. Osteoclast Cytotoxicity Assay BMMs were plated at a density of 1 × 104 cells/well on 96-well plates in triplicate. After treatment with M-CSF (30 ng/mL) and austalide K, cells were incubated for 3 days. Cell viability was measured by using Cell Counting Kit-8 (CCK-8; Dojindo Molecular 256 Technologies, Kumamoto, Japan) according to the manufacturer’s protocol. This seems to be BMM cytotoxicity, not osteoclast as they did not use RANKL for inducing differentiation. Please include the data for osteoclast cytotoxicity too. 11. I am surprised to see that no Supporting information was mentioned or cited in the text. What is the reason? The whole study is based on the successful isolation, identification of the compound austalide K. So the related data should be included in the main manuscript. 12. Figure S7: what is another band below NFATC1?

Author Response

Review 1 comment

This is an incredible research article in bone research and may open new hope for the treatment of bone diseases by targeting bone resorption and formation simultaneously, which may lead to potential drug development. This research and manuscript design is very comprehensive and appropriate for the International Journal of Molecular Sciences. However, I have the following suggestions:

  1. Please increase the size of Figure 2A, 4A and mark the cells for clarity. 2. Line 128, levels of NFATc1, TRAP. DC-STAMP; please replace with comma. 3. Figure 4C, please keep the X-axis name at the middle of the corresponding column bar. 4. Figure 5 (middle one), XY-axis color is different. 5. Please use italic wherever required, for example, line 158, in vivo, etc. 6. Please use the appropriate symbols, for example, line 23, nuclear factor-kB ligand. It should be κ. 7. Line 181, inhibiting NFAT c1 expression; please delete the space. (Ans) As you advised, I modified it.
  2. Line 195, Taken together, austalide K may lead to novel drugs for treating osteoporosis. I think it’s better to write may lead to the development or discovery or something like that. Also, check line 304. (Ans) According to your advice, I modified ‘novel drugs for treating osteoporosis’ to ‘the development for novel drugs for treating osteoporosis’ in the discussion section.
  3. In the material and method section, please mention how Bone marrow-derived macrophages cells were collected, separated. (Ans) I added some additional sentences about the collection method of macrophages cells in the material and method section.
  4. 4.6. Osteoclast Cytotoxicity Assay BMMs were plated at a density of 1 × 104 cells/well on 96-well plates in triplicate. After treatment with M-CSF (30 ng/mL) and austalide K, cells were incubated for 3 days. Cell viability was measured by using Cell Counting Kit-8 (CCK-8; Dojindo Molecular 256 Technologies, Kumamoto, Japan) according to the manufacturer’s protocol. This seems to be BMM cytotoxicity, not osteoclast as they did not use RANKL for inducing differentiation. Please include the data for osteoclast cytotoxicity too. (Ans) As you advised, the survival rate (%) of RANKL-induced BMM with the treatment of austalide K was investigated, and we added the Figure S8 in the supplementary information.
  5. I am surprised to see that no Supporting information was mentioned or cited in the text. What is the reason? The whole study is based on the successful isolation, identification of the compound austalide K. So the related data should be included in the main manuscript. (Ans) The manuscript will be too massive if the supplementary information will be transferred to the main body of manuscript. So, we uploaded the pivotal results and the summary in the manuscript, and then describes the detail information about the extracts of austalide K in the supplementary information.
  6. Figure S7: what is another band below NFATC1? (Ans) I assume the band is non-specific band. Sometimes we could see some non-specific band in western blots. It is a common practice. Although there is a non-specific band, the western band of NFATc1 has correct molecular size. That means the antigen-antibody of NFATc1 binding was not bad.

Reviewer 2 Report

Kwang-Jin Kim et al. reports that austalide K, isolated from the marine fungus Penicillium rudallenes, has dual activities in bone remodeling. After close evaluation of manuscript I suggest revision according to the next points:

  1. The title is too long, I suggest deleting "A Marine Natural Product," from the title.
  2. In Keywords: please remove double term osteoclast; osteoblast.
  3. All abbreviations should be clarified when first time mentioned.
  4. The introduction is focused in osteoporosis, while austalide K is not characterized. The introduction should provide more information about marine derived compound.
  5. The Chemical Structure of Austalide K is presented in Results section. Was this structure ifentified by authors? To which class of chemical compounds is belongs Austalide K?
  6. The Fig. 2D is not informative. It could be excuded.
  7. In legend to Fig.4CIn legend to Fig.5: where are : I don't see "NC and PC". What it mean?
  8. Discussion is weak. Please compare results of your study with other recently published.
  9. The pharmacokinetic plays an important role, Please address this issue. What about the pharmacokinetic of  Austalide K?
  10. The references are not numbered. It is impossible to check correctness of citations in the text.

Author Response

Review 2 comment

Kwang-Jin Kim et al. reports that austalide K, isolated from the marine fungus Penicillium rudallenes, has dual activities in bone remodeling. After close evaluation of manuscript I suggest revision according to the next points:

  1. The title is too long, I suggest deleting "A Marine Natural Product," from the title. 2. In Keywords: please remove double term osteoclast; osteoblast. All abbreviations should be clarified when first time mentioned. (Ans) As you advised, I modified it.
  2. The introduction is focused in osteoporosis, while austalide K is not characterized. The introduction should provide more information about marine derived compound. (Ans) As you advised, I added the additional information in the introduction section.
  3. The Chemical Structure of Austalide K is presented in Results section. Was this structure ifentified by authors? To which class of chemical compounds is belongs Austalide K? (Ans) We collected marine natural products from sea, and then selected osteoporosis-related substances through in vitro screening. After that, the extraction process and contents are explained in material and method and Result section.
  4. The Fig. 2D is not informative. It could be excuded. (Ans) The meaning of Fig.2D is confirmation that there is no cytotoxicity of austalide K itself on the BMMs. So, we think the experiment like Fig.2D on the cytotoxicity of austalide K is meaningful.
  5. In legend to Fig.4C. In legend to Fig.5: where are : I don't see "NC and PC". What it mean? (Ans) As you advised, I modified it.
  6. Discussion is weak. Please compare results of your study with other recently published. (Ans) As recommended, some sentences are added in the discussion section.
  7. The pharmacokinetic plays an important role, Please address this issue. What about the pharmacokinetic of Austalide K? (Ans) We added some sentences like below in the discussion section. The pharmacokinetic plays an important role in the progress of drug development. Austalide K didn't give satisfactory bioavailability although it showed profound in vivo activity. Characterization and biological activity of the major metabolites of austalide K is in progress.
  8. The references are not numbered. It is impossible to check correctness of citations in the text. (Ans) As you advised, I modified it.

Round 2

Reviewer 1 Report

Now the manuscript looks good. However, I have following comments:

  1. Italicize wherever required, for example line 100, in vitro and in vivo.
  2. Line 292, 304, BMMs, but line 296, BMCs. Please check and confirm.

Author Response

  1. Italicizewherever required, for example line 100, in vitro and in vivo.  (Answer) As you advised, I modified it.
  2. Line 292, 304, BMMs, but line 296, BMCs. Please check and confirm. (Answer) The BMM of line292 was modified to BMC. But I think the BMM on line304 is correct.

Reviewer 2 Report

The paper was revised and improved. However some issues still requare additional attention.

  1. The phrase "...The pharmacokinetic plays an important role in the progress of drug development...." require reference support (see for example one of the recent: https://doi.org/10.3390/md18110557)
  2. The discussion is ill weak. I don't find comparison of results of current study with others.

Author Response

  1. The phrase "...The pharmacokinetic plays an important role in the progress of drug development...." require reference support (see for example one of the recent: https: //doi.org/10.3390/md18110557) (Answer) The related reference was cited as recommended.
  2. The discussion is ill weak. I don't find comparison of results of current study with others.  (Answer)  As you advised, we supplemented the discussion and compared it to other studies.

Round 3

Reviewer 2 Report

1. For my comment: The phrase "...The pharmacokinetic plays an important role in the progress of drug development...." require reference support (see for example one of the recent: https://doi.org/10.3390/md18110557) authors provide response that "(Answer) The related reference was cited as recommended". However, the citation was not included in the text.

Author Response

  1. For my comment: The phrase "...The pharmacokinetic plays an important role in the progress of drug development...." require reference support (see for example one of the recent: https://doi.org/10.3390/md18110557) authors provide response that "(Answer) The related reference was cited as recommended". However, the citation was not included in the text.

 (Answer) We apologize for the error. As your advice, I have added the reference [29] in discussion.

Round 4

Reviewer 2 Report

Accept